# Health Effects of Drinking Water Produced from Deep Sea Water: A Randomized Double-Blind Controlled Trial

**DOI:** 10.3390/nu14030581

**Published:** 2022-01-28

**Authors:** Hiroaki Takeuchi, Yu Yoshikane, Hirotsugu Takenaka, Asako Kimura, Jahirul Md. Islam, Reimi Matsuda, Aoi Okamoto, Yusuke Hashimoto, Rie Yano, Koichi Yamaguchi, Shouichi Sato, Satoshi Ishizuka

**Affiliations:** 1Department of Medical Laboratory Sciences, Health and Sciences, International University of Health and Welfare Graduate School, 4-3 Kouzunomori, Narita-City 286-8686, Chiba, Japan; a-kimura@iuhw.ac.jp (A.K.); 21s3057@g.iuhw.ac.jp (J.M.I.); 1857070@g.iuhw.ac.jp (R.M.); 1857025@g.iuhw.ac.jp (A.O.); y.hashimoto@iuhw.ac.jp (Y.H.); 21s1119@g.iuhw.ac.jp (R.Y.); yamaguchi51@iuhw.ac.jp (K.Y.); s-shouichi@iuhw.ac.jp (S.S.); 2Department of Human Living Sciences, Notre Dame Seishin University, 2-16-9 Ifuku-cho, Kita-ku, Okayama-city 700-8516, Okayama, Japan; yyoshikane@m.ndsu.ac.jp; 3DyDo-T Beverage Co. Ltd., 1310-1 Hanechou-ko, Muroto-City 781-6741, Kochi, Japan; takenaka@dt-beverage.com; 4Center for Regional Sustainability and Innovation, Kochi University, 2-17-47 Asakurahonmachi, Kochi-City 780-8073, Kochi, Japan; zuka@kochi-u.ac.jp

**Keywords:** health effect, deep sea water (DSW)-based drinking water, body maintenance, short-chain-fatty-acid, sIgA, daidzein-to-equol conversion, intestinal microbiota

## Abstract

Global trends focus on a balanced intake of foods and beverages to maintain health. Drinking water (MIU; hardness = 88) produced from deep sea water (DSW) collected offshore of Muroto, Japan, is considered healthy. We previously reported that the DSW-based drinking water (RDSW; hardness = 1000) improved human gut health. The aim of this randomized double-blind controlled trial was to assess the effects of MIU on human health. Volunteers were assigned to MIU (*n* = 41) or mineral water (control) groups (*n* = 41). Participants consumed 1 L of either water type daily for 12 weeks. A self-administered questionnaire was administered, and stool and urine samples were collected throughout the intervention. We measured the fecal biomarkers of nine short-chain fatty acids (SCFAs) and secretory immunoglobulin A (sIgA), as well as urinary isoflavones. In the MIU group, concentrations of three major SCFAs and sIgA increased postintervention. MIU intake significantly affected one SCFA (butyric acid). The metabolic efficiency of daidzein-to-equol conversion was significantly higher in the MIU group than in the control group throughout the intervention. MIU intake reflected the intestinal environment through increased production of three major SCFAs and sIgA, and accelerated daidzein-to-equol metabolic conversion, suggesting the beneficial health effects of MIU.

## 1. Introduction

The utilization of deep sea water (DSW) has expanded to the energy, agriculture, food, cosmetics, and public health fields [1]. DSW obtained from depths of >200 m is characterized by high purity, stability at low temperatures, high mineral concentrations, and the presence of bioactive nutritional species [2]. Bottled commercial DSW-based drinking water produced by different methods such as desalinization, is currently available on the market; this commercial product is gaining popularity due to its potential benefits to human health, as confirmed by various animal studies [3,4,5,6]. However, clinical trials are required to clarify the safety and validity of the effects of DSW-based drinking water on human health.

Previous clinical trials using DSW-based drinking water have confirmed that DSW-based drinking water (RDSW; hardness, 1000 mg/L of Ca/Mg) has various beneficial effects on human health, for example with regard to hemorheology, allergies, immunology, infectious diseases (e.g., anti-*Helicobacter pylori* activity), and the intestinal environment [7,8,9,10,11,12]. For example, a recent clinical trial reported that drinking RDSW improved human health due to the increased production of short-chain fatty acids (SCFAs) in the intestinal environment and urinary isoflavones [12].

The intestinal environment comprises the microbiota, microbiota-derived metabolites, and ingesta, and includes the microbe–microbe and host–microbe interactions, which play a fundamental role in human health [13]. Healthy foods and beverages, including probiotics and supplements, are widely consumed to maintain and support the intestinal environment and microbiota [14,15,16,17]. Recently, fecal microbiota transplantation has been included in the treatment of autoimmune diseases, hepatitis, metabolic syndromes, and mental disorders [18,19,20,21] via modulation of the gut microbiota [22,23]. Gut microbes produce various metabolites, such as isoflavones, that are beneficial to human health [24,25,26]. Isoflavone and other metabolite contents vary among individuals due to differences in the intestinal environment, including microbial identity and activity, stability, and variations in the concentrations of endogenous compounds that modulate biotransformation pathways [27,28]. Equol, which is produced from daidzein by gut microbes, is one of the most physiologically active isoflavones. However, only 30–50% of the human population produces equol; a regional difference exists due to the frequency of soybean consumption. Current research has focused on manipulating the gut environment to enhance equol production.

The purpose of the present study was to assess whether DSW-based drinking water (MIU; hardness, 88) could modulate intestinal microbe biomarkers in healthy adults.

## 2. Materials and Methods

### 2.1. Clinical Study Design

This randomized double-blind controlled trial was designed to compare the intestinal environment of individuals in response to drinking MIU vs. mineral water using a self-administered questionnaire and stool and urine sample analysis. The study was conducted in Muroto, Kochi, Japan, from 2018 to 2020. The study protocol, although severely restricted in terms of time and budget, was approved by the Ethics Committees of Kochi University (approval no. 28–93) and the International University of Health and Welfare (approval no. 18-lo-100) and was conducted in accordance with the ethical standards described in the 1964 Declaration of Helsinki and its later amendments. The questionnaire and stool and urine samples were collected before and after the intervention.

### 2.2. Participants

The study cohort included 114 healthy adults residing in Muroto, Kochi, Japan, who agreed to participate and submitted a signed consent form (Figure 1). Potential participants with any current illness, those using any prescription or commercial drugs or dietary supplements, and pregnant women were excluded from the study. Of the 107 healthy adults who met the inclusion criteria, 82 who correctly completed the questionnaire were randomly divided into 2 groups: the MIU group (*n* = 41) and the mineral water (control) group (*n* = 41). The characteristics of the study participants are presented in Table 1. There were no significant differences in terms of age, sex, body mass index, and biomarker concentrations between the 2 groups (Mann–Whitney *U* test).

### 2.3. Ingestion Schedule

The study participants in the MIU group consumed bottled MIU water (Dydo-miu; hardness, 88; Dydo-Takenaka Beverage Co., Ltd., Kochi, Japan) (Appendix A), whereas those in the control group consumed mineral water (hardness, 0–20). The most popular mineral water consumed in Japan was used. Neither of the types of water had calories, proteins, fats, carbohydrates, or vitamins. Both bottled waters were commercially available in Japan and the labels were changed to mask the type of water. The participants were instructed to consume 1 L of the assigned water type daily for 12 weeks.

### 2.4. Evaluation

A self-administered questionnaire was implemented to assess the general health status of the participants. We analyzed the following fecal biomarkers: secretory immunoglobulin A (sIgA), 5 putrefactive products (phenol, *p*-cresol, 4-ethylphenol, indole, and skatole), and 9 SCFAs (succinic, lactic, formic, acetic, propionic, isobutyric, butyric, 3-methylbutanoic, and valeric acids) at TechnoSuruga Laboratory Co., Ltd. (Shizuoka, Japan) [29,30]. Three urinary isoflavones (genistein, daidzein, and equol) were measured in the urine samples. Nonparametric analysis was conducted to assess the differences in these biomarkers before and after the intervention. Based on the changes in the biomarker concentrations throughout the intervention period, multiple logistic regression analysis was performed to evaluate the relationship between the water type and biomarkers.

### 2.5. Self-Administered Questionnaire

A total of 86 participants, including 4 who did not submit urine and fecal samples, answered the questions regarding general gut health and eating habits (i.e., constipation (evacuation frequency, incomplete evacuation, straining at stool, dyschezia, etc.), abdominal discomfort, medication use, and consumption of unusual foods and beverages). Constipation was defined in accordance with the guidelines of the World Gastroenterology Organization [31].

### 2.6. Measurement of Fecal and Urine Samples

Measurements of the samples were taken as previously described [12]. Fecal sample analyses were performed at TechnoSuruga Laboratory Co., Ltd. [29,30].

#### 2.6.1. Fecal sIgA

Here, 0.1 g of each fecal sample suspended in a mixture containing 0.1 mM perchloric acid and 3% phenol was heated and vortexed, followed by centrifugation (15,350× *g*, 10 min) according to previous protocol [12]. The supernatant was collected and filtered (pore size, 0.45 µm) to measure sIgA and SCFA contents. sIgA levels were measured using the Human IgA ELISA Quantitation kit (E80–102; Bethyl Laboratories Inc., Montgomery, TX, USA) and a microplate reader (Varioskan Flash; Thermo Fisher Scientific, Waltham, MA, USA).

#### 2.6.2. Fecal Putrefactive Products

For this, 0.1 g of each fecal sample suspended in 2.5 mL of phosphate buffer including 0.4 mg/L of 4-isopropylphenol as an internal standard was employed to the previous procedures [12]. Briefly, 1 mL of the supernatant was dehydrated and purified with 3 cartridges such as sodium sulfate drying cartridge (Bond Elut LRC; Agilent Technologies, Tokyo, Japan), C18 cartridge (Smart SPE C18-30; AiSTI Science, Wakayama, Japan), and PSA cartridge (Smart SPE PSA-30; AiSTI Science).

The levels of indole and phenol were determined by a single quadrupole gas chromatograph–mass spectrometer (QP-2010; Shimadzu, Kyoto, Japan) equipped with a capillary column (Inert cap WAX; GL Science, Tokyo, Japan). Helium was used as the carrier gas. The injector and interface temperatures were maintained at 240 °C and 230 °C, respectively. For the analysis, 1 µL of the extract was subjected to the splitless mode. The mass spectrometer was operated in the electron impact ionization mode at 70 eV. The measurements were recorded, and data were obtained from the selected ion-monitoring mode for quantification.

#### 2.6.3. Analysis of Intestinal Microbiota

The fecal samples suspended in a buffer containing 4 M guanidium thiocyanate, 100 mM Tris-HCl, and 40 mM ethylenediaminetetraacetic acid were pulverized as previously described [12]. Following this, DNA was extracted from the suspension using the Magtration System 12GC and GC series MagDEA DNA 200 (Precision System Science Co., Ltd., Matsudo, Japan). The final DNA concentration (10 ng/μL) was subjected to the analysis of the microbial community structure by terminal restriction fragment length polymorphism and next-generation sequencing using the MiSeq system (Illumina, San Diego, CA, USA) at TechnoSuruga Laboratory Co., Ltd. [29,30,32]. Bioinformatic analysis was performed using the Ribosomal Database Project (RDP) Multiclassifier tool and Metagenome@KIN software (World Fusion Co., Ltd., Tokyo, Japan) based on data from bacterial species as determined by RDP taxonomic analysis.

#### 2.6.4. Urinary Isoflavones

Measurement of urinary isoflavones (genistein, daidzein, and equol) was performed according to the previous procedures [12]. Briefly, the mixture containing 800 µL urine, 80 µL 1 M sodium acetate and 8 µL β-glucuronidase/sulfatase solution was hydrolyzed, followed by addition of 80 µL of propyl 4-hydroxybenzoate as an internal standard, and then the analytes were extracted. A 20 µL of the residue dissolved in 400 µL of methanol was subjected to a high-performance liquid chromatography system (Shimadzu Co., Ltd., Koto, Japan) and evaluated under the previous conditions [12]. The detection limit of urinary isoflavones was as follows; 100 ng/mL for genistein and daidzein, and 200 ng/mL for equol. The urinary isoflavones were corrected with urinary creatinine (expressed as mg/g-Cre). The equol was corrected for the presence of daidzein (equol/daidzein expressed as g/g-D). The metabolic efficiency of daidzein-to-equol conversion was calculated as equol/equol + daidzein (expressed as g/g-E + D) [33].

### 2.7. Statistical Analysis

Throughout the study, we basically performed statistical analysis with nonparametric analyses unless otherwise indicated whenever necessary. The normality test was performed using the Kolmogorov–Smirnov method to assess the normal distribution. Differences in the preintervention biomarker concentrations between the MIU and mineral water (control) groups were identified by the Mann–Whitney U test. There were no significant differences in the baseline characteristics of the participants among the two groups (Table 1). The measured values of all biomarkers of the 2 groups before and after the intervention were compared using the Wilcoxon signed-rank test (*p* < 0.05) as appropriate (Table 2). Fecal formic acid was excluded from the statistical analysis due to the limited number of samples. Based on the differences in the changes to fecal biomarker concentrations throughout the intervention period, multiple logistic regression analysis was performed to evaluate the relationship between the water types and fecal biomarkers (*p* < 0.05). Multiple logistic regression analysis was performed using adequate data excluding extreme values. Participants with detectable (≥ 200 ng/mL) and undetectable equol levels were classified as equol producers and equol nonproducers, respectively. Equol nonproducers were excluded from the statistical analysis for equol level assessment. Daidzein-to-equol conversion efficiency and relative abundance of equol-producing bacteria detected in the equol producers were assessed using the Wilcoxon signed-rank test. All analyses were performed with BellCurve for Excel ver. 3.20 (Social Survey Research Information Co., Ltd., Tokyo, Japan).

## 3. Results

### 3.1. Self-Administered Questionnaire

The questionnaire (*n =* 86) revealed that three and two participants in the MIU and control groups, respectively, suffered from constipation as per the guidelines of the World Gastroenterology Organization prior to the study [31]. Drinking water ameliorated the symptoms of constipation in all (100%) and none (0%) of the individuals in the MIU and control groups, respectively. Improvement in constipation was observed only in the MIU group, although the small number of samples limited the suitability of the definition of constipation.

### 3.2. Analysis of Fecal sIgA, Putrefactive Products, and SCFAs

The values of fecal biomarkers throughout the intervention period are summarized in Table 2. Overall, in the postintervention period, the sIgA concentration increased in the MIU group and decreased in the control group compared with the concentrations in preintervention period. Thus, the preintervention data of the two subgroups were analyzed in detail to evaluate the sustainable effect on intestinal immune status between the low and high sIgA subgroups with the median (<500 and ≥500 µg/g, respectively). The postintervention sIgA concentration significantly increased in both low-value subgroups. Conversely, the sIgA concentration significantly decreased in the high-value subgroup of the control group but remained unchanged in the high-value subgroup of the MIU group(Figure 2). Among putrefactive products in the postintervention period, indol significantly increased in the MIU group and phenol significantly increased in the control group.

Differences in the changes in SCFAs concentrations throughout the intervention period were analyzed using the Mann–Whitney U test (Table 2). In few minor SCFAs, an increase/decrease of amounts was observed. In particular, the levels of succinic acid and lactic acid decreased in the postintervention period in the MIU group but not in the control group. However, overall, the differences in the changes in the nine SCFAs among the two groups were similar. On the other hand, the total amounts of the three major SCFAs (acetic, propionic, and butyric acids) slightly increased during the postintervention period in the MIU group. A decrease was only observed in the control group (*p* < 0.1, Wilcoxon signed-rank test), with a 23% difference between the two groups (Figure 3a). The populations of the responders whose concentrations of three major SCFAs increased in the postintervention period were significantly higher in the MIU group than in the control group, as determined by the Chi-squared test (Figure 3b). Notably, there were no significant differences between males and females.

Based on the differences in the changes to the fecal biomarker concentrations throughout the intervention period, multiple logistic regression analysis was performed to evaluate the relationship between the two types of water and fecal biomarkers. The results revealed that MIU significantly affected only one biomarker (butyric acid). In addition, MIU more noticeably impacted the intestinal concentrations of the three major SCFAs. Among the 82 participants, formic acid was detected in the range of 0.01–0.02 mg/mL (limit of detection, 0.01 mg/mL) in relatively few samples, suggesting that only minute amounts of formic acid are produced in the human intestine.

### 3.3. Analysis of Urinary Isoflavones

The results of urinary isoflavone analysis of the 41 and 42 participants in the MIU and control groups, respectively, are presented in Figure 1 and Table 1. The focus of this analysis was the differences in the changes to equol concentrations, which is among the most physiologically active isoflavones [34]. Throughout the intervention period, urinary equol was detected in 20 and 12 participants (who were identified as equol producers) in the MIU and control groups, respectively. Overall, equol was identified in 38.6% (32/83) of participants. Interestingly, among the 32 equol producers, 8 participants (6 in the MIU group and 2 in the control group, respectively) became equol producers during the intervention period.

If urinary isoflavones were not detected in a sample, the value was considered 0. The equol value was corrected with creatinine (mg/g-Cre) and daidzein (g/g-D). In addition, the metabolic efficiency of daidzein-to-equol conversion was calculated as equol/equol + daidzein (g/g-E + D), as mentioned above. The changes in equol concentrations throughout the intervention period between the two groups are presented in Table 2. All three evaluations revealed increased equol concentrations in the MIU group. The metabolic efficiency of daidzein-to-equol conversion significantly increased in the MIU group compared with the control group (*p* < 0.1, Wilcoxon) (Figure 4).

### 3.4. Analysis of Fecal Microbiota in Equol Producers

Fecal microbiota analysis of the 32 equol producers identified 15 equol-producing bacteria (Table 3) [34,35]. The metabolic efficiency of daidzein-to-equol conversion was significantly greater in the equol producers in the MIU group as compared to the control group throughout the intervention period. Thus, relative abundance of equol-producing bacteria detected in the equol producers was compared before and after drinking MIU. Of 15 equol-producing bacteria, the median of relative abundance of *Bacteroides ovatus* especially increased from 0.064% (IQR, 0.021–0.262%) to 0.126% (0.034–0.217%) without statistical significance (Wilcoxon signed-rank test).

Furthermore, the differences in intestinal microbes were analyzed, except for 15 known equol-producing bacteria in the 8 participants who became equol producers during the intervention period. Intestinal microbiota analysis was able to be performed in 5 of the 8 participants (4 in the MIU group and 1 in the control group, respectively). The results revealed that the relative abundances of *Blautia wexlerae* and *Streptococcus cristatus* increased throughout the intervention period in the five equol producers (Table 4).

## 4. Discussion

Current world trends are focused on the balanced intake of foods and beverages to promote human health. However, the quality of products currently available in the market is questionable because of a lack of clinical studies. RDSW (hardness, 1000) is reported to improve the intestinal environment [12]. In this study, the average of total amount of the three SCFAs slightly increased in MIU (hardness, 88); however, MIU mainly increased sIgA production as a fecal biomarker. It also improved the metabolic efficiency of daidzein-to-equol conversion and, subsequently, the intestinal environment. Intriguingly, MIU significantly induced sIgA secretion, and the increased level was continuously maintained irrespective of the sIgA level in the preintervention period. However, this was not seen in the control group, indicating that MIU sustainably maintains the intestinal immune status with inducible sIgA. Furthermore, the metabolic efficiency of daidzein-to-equol conversion was accelerated in the MIU group. These findings were not observed in a previous clinical study with RDSW [12]. We previously found increased concentrations of five SCFAs in the RDSW. Thus, the influences differed even with similar DSW-based drinking water, which was likely due to the hardness and manufacturing process. The reference values of the constituents (i.e., putrefaction, SCFA, isoflavones) measured in this study are not defined at present. New investigations could assess these effects on immune and inflammatory responses, gut microbiota, and microbial products in healthy adults. However, at least in healthy persons, the increased constituents (IgA and SCFAs) are considered as beneficial to the body and not a disadvantage. We found no adverse events during this clinical study.

It is generally accepted that isoflavones can ameliorate the symptoms of various syndromes and diseases, including cancers. In particular, the physiological activities of equol produced by bacterial daidzein conversion in the intestine can reduce the risk of several diseases [34]. Epidemiological evidence suggests that equol production (~30–50% worldwide) is related to environmental factors, dietary habits, and gut microbes that convert daidzein to equol (daidzein-to-equol conversion). To date, a limited number of bacteria capable of daidzein-to-equol conversion have been identified [34,35]. Thus, further studies are warranted to identify foods, beverages, and as-yet unidentified bacteria capable of daidzein-to-equol conversion to improve the intestinal environment and maintain human health.

In this study, MIU increased the metabolic efficiency of daidzein-to-equol conversion, which benefited human health by improving the intestinal environment. Of the 15 equol-producing bacteria detected in 32 subjects, the relative abundance of *B. ovatus* especially increased in the MIU group without statistical significance, suggesting that MIU intake influenced the equol-producing bacteria, including *B. ovatus*. Furthermore, fecal microbiota analysis of five participants who became equol producers throughout the intervention period demonstrated that the proportions of *B. wexlerae* and *S. cristatus* had significantly increased in the postintervention period. *Blautia* spp., which is dominant in the human intestine, can ameliorate the symptoms of inflammatory and metabolic diseases and improve antibacterial activity [36,37]. These physiological activities of probiotics are beneficial to human health [38]. In particular, *B*. *wexlerae* reduces inflammation associated with obesity-related complications and produces acetic acid as a final product of glucose fermentation [38,39]. The relative abundance of *B*. *wexlerae* in the postintervention period was observed in all five equol producers, suggesting that *B*. *wexlerae* could be involved in equol production. However, no study has investigated the relationship between equol production and *Blautia* spp. in the intestine. Hence, further studies are needed to investigate the contribution of *B*. *wexlerae* to equol production and/or the metabolic efficiency of daidzein-to-equol conversion. Previous studies have reported that *S. cristatus* peptides repressed expression of the virulence genes of *Porphyromonas gingivalis*, which had an inhibitory effect on the oral microbiota [40,41]. The physiological function of *S. cristatus* is mostly unknown; thus, in vitro studies are warranted to investigate the activities of *S. cristatus* in the intestinal environment to clarify the connection with equol. In addition, to evaluate the improvement of intestinal microbiota, all 82 fecal samples collected during the preintervention period were subjected to PCR analysis with specific primers for the amplification of methicillin-resistant *Staphylococcus aureus* (MRSA) [42]. The results demonstrated that three participants in the MIU group were healthy MRSA-carriers and all three were free of MRSA postintervention, indicating that MIU eventually cleared MRSA from the intestine, although the results were limited by the small number of MRSA carriers (data not shown). Furthermore, an increased concentration of indol was observed postintervention in the MIU group. Indol is a metabolite produced from tryptophan by intestinal microbiota, and this thus suggests that MIU intake influenced the intestinal microbiota/condition.

The health effects of isoflavones, including equol, are dependent on the quantity and bioavailability of absorbed nutrients [43]. Isoflavones are converted to aglycones (daidzein and genistein) from glycones (daidzin and genistin) by enzymes (i.e., β-glucosidase) in intestinal microbes/conditions. The aglycones absorbed in the body physiological function. Thus, we measured urinary isoflavones (daidzein, genistein, and equol) to evaluate the health effects. The lifestyle factors of the participants were strictly monitored to assess dietary habits and the consumption of unusual foods, beverages, and supplements throughout the study period. MIU probably influenced not only the metabolic efficiency of daidzein-to-equol conversion but also the absorption efficiency via the intestinal environment. An improvement of constipation was suggested in the MIU group only, although no significant difference was observed due to the small number of participants fitting the definition of constipation. Taken together, the findings of this study indicate that MIU intake induces these biomarkers in the intestinal environment.

## 5. Conclusions

This clinical study revealed that a long-term intervention with MIU mainly induced sIgA production and increased the metabolic conversion of daidzein-to-equol, suggesting an adaptation of the host/microbe which influences human health.

## Figures and Tables

**Figure 1 nutrients-14-00581-f001:**
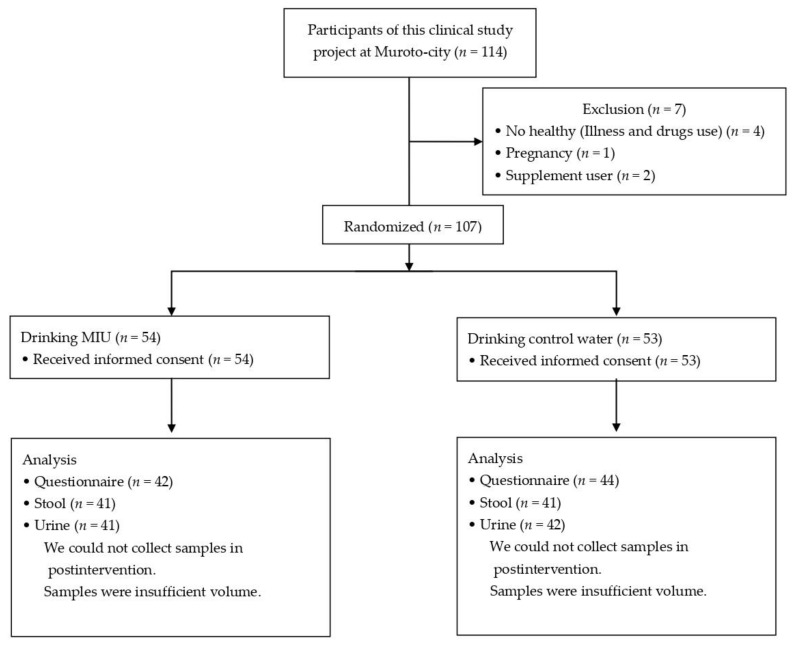
Flow diagram of this clinical study. A total 107 healthy adults were enrolled from Muroto, Kochi, Japan. Potential participants with any current illness, those using any prescription or commercial drugs or dietary supplements, and pregnant women were excluded. Participants in the experimental group consumed MIU (hardness, 88) and those in the control group consumed mineral water (hardness, 0–20).

**Figure 2 nutrients-14-00581-f002:**
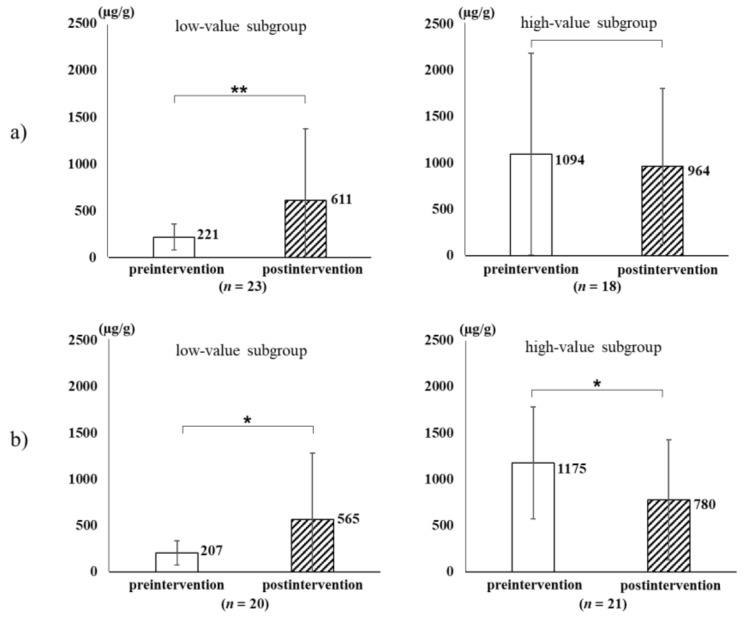
Differences in the changes to sIgA concentrations throughout the intervention period in the MIU (**a**) and control (**b**) groups. Participants in the MIU and control groups was further classified into 2 subgroups: low and high sIgA preintervention levels (<500 vs. ≥500 µg/g). The concentration of sIgA throughout the intervention period significantly increased in both low-value subgroups irrespective of the water type. However, sIgA significantly decreased in the high-value subgroup of the control group but remained unchanged in the MIU group. Open bar, preintervention; hatched bar, postintervention. Bar depicts standard deviation. * *p* < 0.05; ** *p* < 0.01.

**Figure 3 nutrients-14-00581-f003:**
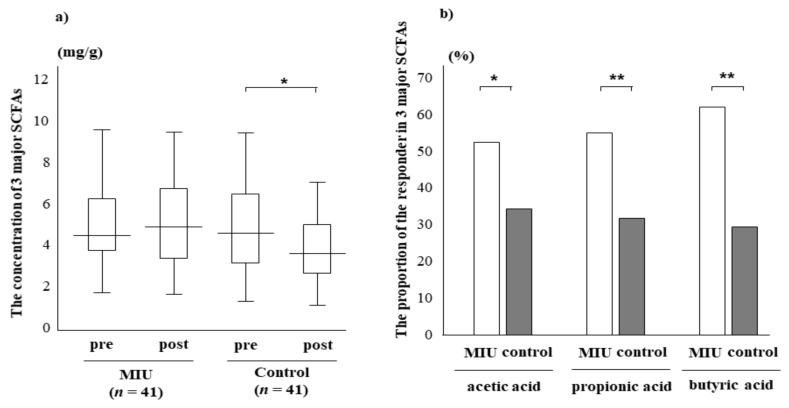
Effect of MIU or control water on fecal biomarker concentrations of 3 SCFAs (acetic acid, propionic acid, and butyric acid) throughout the intervention period. (**a**) The concentrations of the SCFAs decreased in the control group (* *p* < 0.1). There was a 23% difference between the 2 groups. The top and bottom of each box indicate the 25th and 75th percentiles, and the solid line within the box is a median. Whiskers depict the minimum and maximum values. pre, preintervention; post, postintervention, (**b**) The proportions of responders were significantly higher in the MIU group than in the control group. * *p* < 0.05; ** *p* < 0.01.

**Figure 4 nutrients-14-00581-f004:**
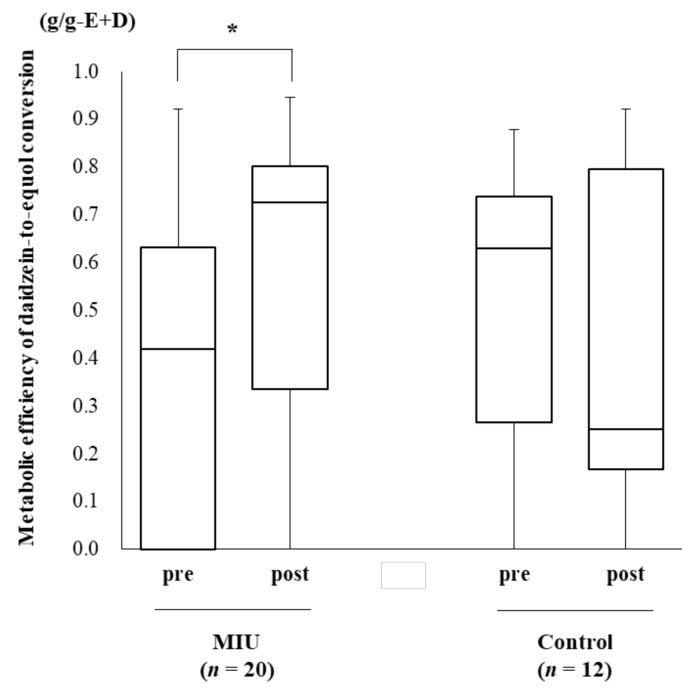
Effect of MIU or control water on the metabolic efficiency of daidzein-to-equol conversion throughout the intervention period. The metabolic efficiency of daidzein-to-equol conversion was significantly prompted in the MIU group. The top and bottom of each box indicate the 25th and 75th percentiles, and the solid line within the box is the median. Whiskers depict the minimum and maximum values. pre, preintervention; post, postintervention, * *p* < 0.1.

**Table 1 nutrients-14-00581-t001:** Preintervention characteristics of the participants from the 2 groups.

	MIU						Mineral Water (Control)					
	Total (*n* = 41)		male (*n* = 17)		female (*n* = 24)		Total (*n* = 41)		male (*n* = 19)		female (*n* = 22)	
**Age (year)**	43	(33–53)	47	(33–52)	42.5	(32–53)	42	(33–57)	37	(37–58)	47	(37–58)
**BMI(Kg/m^2^)**	22.6	(20.7–26.4)	23.5	(22.5–27.2)	21.7	(20.3–23.2)	42	(33–57)	37	(37–58)	47	(37–58)
							22.9	(21.5–25.5)	22.4	(21.3–25.1)	23.2	(21.9–26.4)
**sIgA (μg/g)**	408	(209–651)	394	(202–538)	492	(207–678)	555	(169–1042)	449	(159–1016)	614	(21.9–26.4)

**Putrefaction (μg/g)**												
Phenol	1.3	(0.50–6.70)	4.6	(1.07–13.45)	0.8	(0.45–5.20)	1.4	(0.60–7.20)	2	(0.72–13.55)	1.3	(0.45–5.90)
*p*-Cresol	28.2	(9.15–69.45)	19.7	(6.00–65.50)	40.7	(10.07–81.30)	59.2	(21.20–90.98)	57.4	(23.80–111.42)	60.65	(19.70–78.90)
4-Ethylphenol	2.3	(1.63–4.15)	2.7	N/A	2.3	N/A	1.7	(0.70–2.90)	1.7	(1.50–7.80)	0.7	(0.70–2.47)
Indol	19.4	(11.80–31.75)	19.4	(9.32–32.37)	19.45	(12.60–31.25)	22.8	(11.85–35.90)	30.4	(15.20–41.70)	17.3	(10.00–27.60)
Skatol	2.75	(1.20–7.80)	2.8	(1.27–7.02)	2.7	(0.57–12.12)	4.8	(1.35–10.00)	2.7	(0.09–0.24)	5.8	(1.40–10.00)

**SCFA (mg/g)**												
Succinic acid	0.19	(0.09–0.47)	0.21	(0.11–0.36)	0.16	(0.08–0.50)	0.12	(0.08–0.24)	0.14	N/A	0.11	(0.07–0.24)
Lactic acid	0.19	(0.08–0.68)	0.23	(0.13–0.82)	0.12	(0.07–0.44)	0.11	(0.08–0.17)	0.08	(0.07–0.15)	N/A	
Formic acid	N/A		N/A		N/A		N/A		N/A		N/A	
Acetic acid	3.19	(1.85–4.03)	2.72	(1.65–3.77)	3.27	(2.16–4.04)	2.63	(1.97–3.37)	2.63	(1.90–3.65)	2.59	(1.99–3.69)
Propionic acid	1.02	(0.76–1.28)	1.01	(0.72–1.28)	1.04	(0.79–1.35)	1.12	(0.87–1.57)	1.38	(0.88–1.66)	1.07	(0.85–1.45)
Isobutyric acid	0.16	(0.12–0.20)	0.19	N/A	0.13	(0.11–0.19)	0.15	(0.13–0.19)	0.17	(0.130–0.21)	0.14	(0.130–0.155)
Butyric acid	0.77	(0.54–1.27)	0.73	(0.54–1.05)	0.78	(0.53–1.31)	0.84	(0.54–1.48)	1.03	(0.56–1.80)	0.81	(0.52–1.23)
3-Methylbutanoic acid	0.2	(0.14–0.26)	0.18	(0.14–0.30)	0.21	(0.14–0.25)	0.2	(0.15–0.27)	0.21	(0.16–0.33)	0.2	(0.130–0.25)
Valeric acid	0.21	(0.13–0.31)	0.25	(0.18–0.30)	0.18	(0.12–0.32)	0.17	(0.14–0.29)	0.22	(0.18–0.38)	0.15	(0.13–0.20)

**Urine Isoflavones**												
Daidzein (mg/g-Cre)	0.9	(0.37–2.15)	0.88	(0.57–1.36)	0.92	(0.36–2.70)	0.79	(0.33–1.60)	0.9	(0.48–1.34)	0.76	(0.19–1.71)
Genistein (mg/g-Cre)	1.11	(0.41–2.10)	1.11	(0.41–1.59)	1.12	(0.54–2.41)	1.03	(0.49–1.76)	1.15	(0.75–2.04)	0.87	(0.46–1.43)

	Total (*n* = 20)		male (*n* = 7)		female (*n* = 13)		Total (*n* = 12)		male (*n* = 8)		female (*n* = 4)	
Equol (mg/g-Cre)	0.5	(0.00–1.73)	0.21	(0.00–5.52)	0.57	(0.26–1.52)	0.98	(0.35–2.14)	1.9	(1.08–2.72)	0.32	(0.19–0.48)
Equol (g/g-Da)	0.72	(0.00–1.72)	0.73	(0.00–3.59)	0.72	(0.15–0.88)	1.82	(0.37–3.01)	2.4	(1.06–4.76)	0.31	(0.15–0.93)
Equol (g/g-E + D)	0.42	(0.00–0.63)	0.42	(0.00–0.75)	0.42	(0.13–0.47)	0.63	(0.27–0.74)	0.71	(0.50–0.83)	0.23	(0.13–0.40)

N/A, less than *n* = 6; The data was shown as median and IQR in parentheses; non-parametric analysis (Mann-Whitney U test).

**Table 2 nutrients-14-00581-t002:** The values of fecal biomarkers in the 2 intervention groups.

	MIU (*n* = 41)				Mineral Water (Control) (*n* = 41)			
	Preintervention		Postintervention		Preintervention		Postintervention	
**sIgA (μg/g)**	408	(209–651)	515	(319–1039)	555	(169–1042)	479	(215–893)
**Putrefaction (μg/g)**								
Phenol	1.30	(0.50–6.67)	1.75	(0.85–6.20)	1.40	(0.60–7.20)	1.63	(0.75–4.70) ↑ *
*p*-Cresol	28.19	(9.15–69.45)	42.56	(12.60–95.72)	59.18	(21.20–90.97)	44.80	(18.22–105.77)
4-Ethylphenol	2.32	(1.62–4.15)	1.92	(0.70–2.20)	1.68	(0.70–2.90)	2.24	(0.75–8.45)
Indol	19.36	(11.80–31.75)	21.44	(14.57–41.10) ↑ *	22.79	(11.85–35.90)	18.78	(8.50–28.45)
Skatol	2.76	(1.20–7.80)	1.98	(1.30–6.85)	4.81	(1.35–10.00)	5.50	(2.30–14.75)
**SCFA (mg/g)**								
Succinic acid	0.19	(0.09–0.46)	0.11	(0.080–0.220) ↓ **	0.12	(0.08–0.23)	0.14	(0.085–0.345) ↑ *
Lactic acid	0.19	(0.08–0.68)	0.12	(0.10–0.24) ↓ *	0.11	(0.08–0.17)	0.17	(0.08–0.53)
Formic acid	0.24	N/A	0.18	N/A	0.25	N/A	0.25	N/A
Acetic acid	3.19	(1.85–4.02)	3.00	(1.94–4.21)	2.63	(1.97–3.66)	1.99	(1.72–3.20) ↓ **
Propionic acid	1.02	(0.75–1.28)	1.19	(0.81–1.53)	1.12	(0.87–1.57)	0.99	(0.67–1.26) ↓ **
Isobutyric acid	0.15	(0.12–0.20)	0.15	(0.12–0.20)	0.15	(0.13–0.19)	0.15	(0.12–0.21)
Butyric acid	0.77	(0.54–1.26)	0.89	(0.51–1.11)	0.84	(0.54–1.48)	0.61	(0.33–1.04) ↓ **
3-Methylbutanoic acid	0.19	(0.14–0.26)	0.20	(0.14–0.33)	0.20	(0.15–0.27)	0.25	(0.17–0.34)
Valeric acid	0.21	(0.13–0.31)	0.20	(0.14–0.27)	0.17	(0.14–0.29)	0.21	(0.15–0.32)
**Urine Isoflavones**	(*n* = 20)				(*n* = 12)			
Equol (mg/g-Cre)	0.50	(0.00–1.73)	1.90	(0.50–5.53)	0.98	(0.35–2.14)	1.30	(0.14–2.00)
Equol (g/g-Da)	0.72	(0.00–1.72)	2.74	(0.53–4.06)	1.82	(0.37–3.01)	0.31	(0.00–2.60)
Equol (g/g-E + D)	0.42	(0.00–0.63)	0.73	(0.34–0.80) ↑ #	0.63	(0.27–0.74)	0.25	(0.17–0.80)

*, *p* < 0.05; **, *p* < 0.01; #, *p* < 0.1; N/A, less than *n* = 6; The data was shown as median and IQR in parentheses.; non-parametric analysis (Wilcoxon rank sum test); ↑: increase; ↓; decrease.

**Table 3 nutrients-14-00581-t003:** Detection of equol-producing bacteria in the 32 subjects whose urinary equol levels were detected in this study.

*Adlercreutzia equolifaciens*	*Asaccharobacter celatus*	*Bacteroides ovatus*
*Bifidobacterium animalis*	*Bifidobacterium breve*	*Bifidobacterium longum*
*Finegoldia magna*	*Lactobacillus graminis*	*Lactobacillus intestinalis **
*Lactobacillus mucosae*	*Lactobacillus sakei*	*Pediococcus pentosaceus **
*Slackia equolifaciens*	*Slackia isoflavoniconvertens*	*Streptococcus intermedius*

Twenty and 12 from MIU and control groups, respectively. * Not detected in control group.

**Table 4 nutrients-14-00581-t004:** List of the increased bacteria detected postintervention in 5 equol producers.

*Blautia wexlerae*	*Streptococcus cristatus*	
Increased bacteria detected in 4 of 5 equol producers
*Blautia faecis*	*Butyricicoccus desmolans*	*Clostridium aldenense*
*Clostridium bolteae*	*Eggerthella lenta*	*Enterococcus avium*
*Eubacterium hallii*	*Fusobacterium varium*	*Gemella sanguinis*
*Lactococcus lactis*	*Murimonas intestini*	*Ruminococcus lactaris*
*Solobacterium moorei*

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
