# Peer review of "Health Effects of Drinking Water Produced from Deep Sea Water: A Randomized Double-Blind Controlled Trial"

_nutrients, 2022, doi:10.3390/nu14030581_

Round 1

Reviewer 1 Report

Deep sea water (DSW), which comprehends higher concentrations of Ca, Mg, K, vanadium, and Zn than surface water, has been shown to have positive health effects that need to be further investigated. In this study, the authors repeated a recently published double-blind randomized controlled trial in Nutrients (2020, 12, 2646; doi:10.3390/nu12092646 www.mdpi.com/journal/nutrients; reference 12) using lower hardness in DSW to examine the human intestine health. The article can be improved considering the comments below.

Major concern

i. Appropriate use of statistical methods is of critical importance. If authors do not provide a good knowledge of the statistical, I suggest help from a mathematician/statistician. The statistical part of the experimental study was inconsistent with insufficient explanation.

Data were not represented by measuring central tendency with data dispersion; Mean (average of variables), and median (value located in the middle of the list of variables) are all valid measures of central tendency and need to be correctly added on paper in tables and figures followed by standard deviation (SD) or standard error mean (SEM) for parametric distribution. Likewise, the median and interquartile range (IQR) should be used for non-parametric data. Normality tests has not been described for establish adequate decisions regarding the selection and presentation of statistical methods, which is precious for improve the reliability of the results. As example, the authors used Mann –Whitney U test, which is a non-parametric test. However, it appears that the data are presented as mean and not median. Data in which there is no normal distribution should be presented as median and IQR. In Figure 4 and Table 2, the authors attributed the semiquantitative analysis to urinary isoflavones. Following the key principles of semiquantitative scoring it is not suitable to use parametric analysis for evaluate these data. Multiple logistic regression analysis to assess associations was not demonstrated (table 3 and 4).

ii. Secondary data are all dataset not obtained directly in the research

Why are urinary isoflavone levels considered secondary outcomes? Unlike the questionnaire, in which the answer to this question would obviously depend on one’s research gap, isoflavones, urine biomarker concentrations, and intestinal microbes  (described in line 86) cannot be considered a kind of theory that is being used to explain it, what kind of phenomenon is being analyzed. For me, these parameters are originally intended data.

iii. Gaps that should be discussed

  • Were randomization and blinding considered for group selection?
  • Increased IgA is often necessary for formation of immune complexes that induce the proliferation and secretions of extracellular matrix, cytokines, and chemokines, which result in tissue damage. 
  • Indol, a metabolite produced from tryptophan, active transcription factors that induce the expression of genes such as CYP4501A1, which detoxifies chemicals and toxins. 
  • There are contradictions in the results. Why table 2 did not show any difference for acetic acid, propionic acid, and isobutyric acid in the post-intervention period comparing the MIU group and the control group, but figure 3 shows significant difference in their data. Furthermore, SCFAs which are the main metabolic products of bacterial anaerobic fermentation in the intestine constitute a substantial portion of the essential energy, bidirectionally regulating antigen-specific adaptative immunity. This is very important and deserves to be explained.

iv. Sweeping generalized statements. In the manuscript several unsubstantiated broad "sweeping" statements are made, but they are not supported by rigorous evidence. I would advise that these statements are greatly attenuated to reflect only what is proven in the study and can be accepted.

Examples follow below:

“Bacteroides ovatus significantly affected the conversion efficiency” (line 30-31);

“MIU improved constipation and the intestinal environment through increased production of three major SCFAs, sIgA and daidzein-to-equol metabolic conversion accompanied by B. ovatus” (line 31-32); This is extremely speculative.

…”to evaluate the beneficial effects of MIU for body maintenance” (line 72);

In methodology (sub item 2.4) what does the evaluation refer to? (line 116);

“A self-administered questionnaire was used to assess the general health status of the participants” (line 130-134). This data is few scientific as well as the presented result (line 117-123); a validated questionnaire for the diagnosis and assessment of symptoms of functional constipation should have been used. Among the items, evacuation frequency, incomplete evacuation, obstruction blockade, assessment of abdominal symptoms, discomfort, pain, rectal symptoms…. between others. The reference 31 offers diagnostic criteria, but the instrument used by the authors is not described.

“the health effects differed even with similar DSW-based drinking water…” (line 351)

“Epidemiological evidence suggests that equol production (~30%–50% worldwide) is related to environmental factors, dietary habits, and gut microbes that convert daidzein to equol (daidzein-to-equol conversion). To date, a limited number of bacteria capable of daidzein-to-equol conversion have been identified” (line 356-359). References must be cited.

“Improvement of constipation was confirmed in the MIU group only, although no significant difference was observed due to the small number of participants suitable for a definition of constipation”. (line 396-397)

“… MIU effectively induced the production of three major SCFAs and sIgA, and accelerated daidzein-to-equol metabolic conversion through an increase in the proportion of B. ovatus in the intestine”. (line 402-403) The results of this investigation do not support this conclusion. Authors need to address associations but causality cannot be considered by design experimental defined.

Minor comments

  • Dydo-miu, hardness, 88 (Dydo-Takenaka Beverage Co., Ltd., Kochi, Japan) composition must be presented, even if as a supplementary data.
  • Why line 224-235 is in italic?
  • Were cases of adverse events not reported? Enteritis, a significant change in blood pressure and pulse rate, or other complications were not evaluated and/or related during the study. Biochemical parameters such as BUN, hepatic markers were not performed?
  • SCFAs specification need to be cited in the legend (Figure 3).

Author Response

Reviewer 1

Deep sea water (DSW), which comprehends higher concentrations of Ca, Mg, K, vanadium, and Zn than surface water, has been shown to have positive health effects that need to be further investigated. In this study, the authors repeated a recently published double-blind randomized controlled trial in Nutrients (2020, 12 2646; doi: 10.3390/nu12092646 www.mdpi.com/journal /nutrients; reference 12) using lower hardness in DSW to examine the human intestine health. The article can be improved considering the comments below.

Major concern

â…°.Appropriate use of statistical methods is of critical importance. If authors do not provide a good knowledge of the statistical, I suggest help from a mathematician/statistician. The statistical part of the experimental study was inconsistent with insufficient explanation.

Data were not represented by measuring central tendency with data dispersion; Mean (average of variables), and median (value located in the middle of the list of variables) are all valid measures of central tendency and need to be correctly added on paper in tables and figures followed by standard deviation (SD) or standard error mean (SEM) for parametric distribution.

Likewise, the median and interquartile range (IQR) should be used for non-parametric data. Normality tests has not been described for establish adequate decisions regarding the selection and presentation of statistical methods, which is precious for improve the reliability of the results. As example, the authors used Mann-Whitney U test, which is a non-parametric test. However, it appears that the data are presented as mean and not median. Data in which there is no normal distribution should be presented as median and IQR. In Figure 4 and Table 2, the authors attributed the semiquantitative analysis to urinary isoflavones. Following the key principles of semiquantitative scoring it is not suitable to use parametric analysis for evaluate these data. Multiple logistic regression analysis to assess associations was not demonstrated (table 3 and 4).

Our comments:

>Thank you for your advice and comments. According to your suggestion, we basically used non-parametric analysis (Mann-Whitney and Wilcoxon tests) throughout this study unless otherwise indicated whenever necessary and changed the results from parametric analysis to new results based on non-parametric analysis. We described these in “2.7 statistical analysis” and the adequate positions in the manuscript. We changed to new tables 1 and 2 expressed by median. In Figure 4, we changed to new figure based on non-parametric analysis (Wilcoxon). In addition, we rewrote the figure legend (line 311-315) as follows; “Effect of MIU or control water on the metabolic efficiency of daidzein-to-equol conversion throughout the intervention period. The metabolic efficiency of daidzein-to-equol conversion was significantly prompted in the MIU group. The top and bottom of each box indicate the 25th and the 75th percentiles and the solid line within the box is a median. Whiskers depict the minimum and maximum values. pre, preintervention; post, postintervention, *p < 0.1.”

Regarding to table3, we already performed multiple logistic regression analysis to assess associations and described in “2.7 statistical analysis line206-210” and “3.4 Analysis of Fecal Microbiota in Equol producers line320-322”. We added the sentence in the text (line 322-323) as follows; “This analysis was performed with daidzein-to-equol conversion efficiency of 32 subjects as the object valuable.”

As for table4, we showed the list of increased bacteria detected in postintervention of equol-producer. We did not perform multiple logistic regression analysis due to limited number of available data/participants. However, we believe that the list is helpful and useful for future investigations as candidate bacteria associated with equol-producing. We appreciate for your understanding our thinking.

â…±.Secondary data are all dataset not obtained directly in the research

Why are urinary isoflavone levels considered secondary outcomes? Unlike the questionnaire, in which the answer to this question would obviously depend on one’s reseach gap, isoflavones, urine biomarker concentrations, and intestinal microbes (described in line 86) cannot be considered a kind of theory that is being used to explain it, what kind of phenomenon is being analyzed. For me, these parameters are originally intended data.

Our comments:

>I apologized for your confusion about outcomes. To evaluate the health effects of MIU, we focused a change of intestinal condition using fecal and urine biomarkers such as isoflavone level. It is very important to know/predict the physiological activity of isoflavones in human body as health effect. Isoflavones are converted to aglycones (daidzein and genistein) from glycones (daidzin and genistin) by enzyme (i.e., β-Glucosidase) in intestinal microbes/conditions. The aglycones absorbed in body physiologically function. Thus, we measured urine isoflavones (daidzein, genistein and equol) to evaluate the health effects. We added these into the discussion (line 399-401).    

We changed the urine biomarker from secondary to primary outcomes (abstract line 26-27, 77-79). We appreciate for your understanding our thinking.

â…². Gaps that should be discussed

Were randomization and blinding considered for group selection?

Increased IgA is often necessary for formation of immune complexes that induce the proliferation and secretions of extracellular matrix, cytokines, and chemokines, which result in tissue damage.

Indol, a metabolite produced from tryptophan, active transcription factors that induce the expression of genes such CYP4501A1, which detoxifies chemicals and toxins.

There are contradictions in the results. Why table 2 did not show any difference for acetic acid, propionic acid, and isobutyric acid in the post- intervention period comparing the MIU group and the control group, but figure 3 shows significant difference in their data. Furthermore, SCFAs which are the main metabolic products of bacterial anaerobic fermentation in the intestine constitute a substantial portion of the essential energy, bidirectionally regulating antigen-specific adaptative immunity.

This is very important and deserves to be explained.

Our comments:

>Thank you for your constructive comments.

At first, we planned cross-over trial, however, this clinical study was severely restricted in terms of time and budget, was approved by the Ethics Committees as feasible study. This was added in “2.1 Clinical study design (line 69-70)”.

We agree with your comments and understand these physiological functions. There are huge reports concerned with ambivalent effects in body which is controversial. Because the reference values of these items are not defined yet at present time. At least in healthy person, the increased constituents (IgA and SCFAs) are considered as benefit in body but not disadvantage. For example, considerable increased IgA in infant through breast milk plays an important role as immunoprotection. I think that we need to identify the reference value to distinguish health/benefit from abnormal/diseases/disadvantage. We added these in “Discussion” as follows; The reference values of the constituents (i.e., putrefaction, SCFA, isoflavones) measured in this study are not defined yet at present. Thus, we need to identify the reference values to distinguish benefit from disadvantage and precisely evaluate these effects in body. However, at least in healthy person, the increased constituents (IgA and SCFAs) are considered as benefit in body but not disadvantage (line 363-367).

We appreciate for your understanding our thinking.

In this study, we focused major SCFAs among SCFAs, showed all data and described these. However, according to your suggestion, we described the fact of three constituents (two minor SCFAs, indol and phenol) with statistical differences between the intervention as follows; “In few minor SCFAs, increase/decrease of amount was observed, in particularly, succinic acid and lactic acid decreased in postintervention period in the MIU group but not in control group (line 262-263).”

In addition, we added the sentence of “Among putrefactive products in the postintervention period, indol significantly increased in the MIU group and phenol significantly increased in the control group.” in “3.2 Analysis of Fecal sIgA, Putrefactive Products, and SCFAs (line 231-233).” 

Furthermore, we described “Furthermore, the increased concentration of indol was observed in postintervention in the MIU group. Indol, a metabolite produced from tryptophan by intestinal microbiota, suggesting that MIU intake influenced the intestinal microbiota/condition. (line 395-397).

We apologize for your confusing to table2 and figure3. We changed to new figure3 based on non-parametric analysis according to your suggestion. We rewrote the text as follows; on the other hand, the total amount of the three major SCFAs (acetic, propionic, and butyric acids) increased during the postintervention period in the MIU group and significantly decreased in the control group (Willcoxon, p < 0.1) with a 23% difference between the two groups (Figure 3a). The populations of the responders whose concentrations of three major SCFAs increased in the postintervention period were significantly higher in the MIU group than in the control group (Figure 3b). (line 264-268, 282-288)

â…³. Sweeping generalized statements. In the manuscript several unsubstantiated broad “sweeping” statements are made, but they are not supported by rigorous evidence. I would advise that these statements are greatly attenuated to reflect only what in proven in the study and can be accepted.

Our comments:

>Thank you for your comments and advise. Basically, we rewrote the text according to your suggestions as below.

Examples follow below:

Bacteroides ovatus significantly affected the conversion efficiency” (line 30-31);

>We changed to “The conversion efficiency was associated with abundance of Bacteroides ovatus” in abstract (line 30-31) and in appropriate positions in the text (line 323-324).

“MIU improved constipation and the intestinal environment through increased production of three major SCFAs, slgA and daidzein-to-equol metabolic conversion accompanied by B. ovatus” (line 31-32); This is extremely speculative.

>We changed to “MIU intake reflected the intestinal environment through increased production of three major SCFAs and slgA, and accelerated daidzein-to-equol metabolic conversion,” (line 31-32)

…“to evaluate the beneficial effects of MIU for body maintenance” (line 65);

>We changed to “to evaluate the effects of MIU for body maintenance (line 65)”.

In methodology (sub item 2.4) what does the evaluation refer to? (line 116);

“A self-administered questionnaire was used to assess the general health status of the participants” (line 130-134). This data is few scientific as well as the presented result (line 117-123); a validated questionnaire for the diagnosis and assessment of symptoms of functional constipation should have been used. Among the items, evacuation frequency, incomplete evacuation, obstruction blockade, assessment of abdominal symptoms, discomfort, pain rectal symptoms.... between others. The reference 31 offers diagnostic criteria, but the instrument used by the authors is not described.

>We understand what you mean. Basically, heathy person without abnormal data by periodical clinical laboratory examinations participated in this study. However, we sometimes found the person suitable for a definition of constipation. So, we used a self-administered questionnaire to find the person with the constipation based on ref31. We added few concrete words/examples as definition of constipation as follows; constipation (evacuation frequency, incomplete evacuation, straining at stool, dyschezia, etc.), (line 149-150). In this study, the small number of participants suitable for a definition of constipation was found, thus we omitted the mentions concerned with “constipation” in abstract.

“the health effects differed even with similar DSW-based drinking water...” (line 351)

>We changed to “the influences differed even with similar DSW-based drinking water...” (line 362)”.

“Epidemiological evidence suggests that equol production (~30%-50% world wide) is related to environmental factors, dietary habits, and gut microbes that convert daidzein to equol (daidzein-to-equol conversion). To date, a limited number of bacteria capable of daidzein-to-equol conversion have been identified” (line 356-359). References must be cited.

>The reference was shown [35,36] (line 373).

“Improvement of constipation was confirmed in the MIU group only, although no significant difference was observed due to the small number or participants suitable for a definition of constipation”. (line 396-397)

>we changed to “Improvement of constipation was suggested in the MIU group only, although no significant difference was observed due to the small number or participants suitable for a definition of constipation”. (line 405)” We omitted the mentions concerned with “constipation” in abstract.

“... MIU effectively induced the production of three major SCFAs and slgA, and accelerated daidzein-to-equol metabolic conversion through an increase in the proportion of B. ovatus in the intestine”. (line 402-403) The results of this investigation do not support this conclusion. Authors need to address associations but causality cannot be considered by design experimental defined.

>we changed to “MIU effectively induced the production of three major SCFAs and slgA, and accelerated daidzein-to-equol metabolic conversion. The conversion efficiency was associated with an increase in the proportion of B. ovatus in the intestine” (line 409-410). Likewise, change in abstract. 

Minor comments:

Dydo-miu, hardness, 88 (Dydo-Takenaka Beverage Co., Ltd., Kochi, Japan) composition must be presented, even if as a supplementary data.

>we added it as supplementary data (supplementary table s1, line 133, 539-544).

Why line 224-235 is in italic?

>I could not look the italic (garbling?)… Anyway, I checked all in the manuscript.

Were cases of adverse events not reported? Enteritis, a significant change in blood pressure and pulse rate, or other complications were not evaluated and/or related during the study. Biochemical parameters such as BUN, hepatic markers were not performed?

>I understand what you mean. Eventually, we could perform no more examinations, because this clinical study was severely restricted in terms of time and budget. We described “we found no adverse events through this clinical study.” in the text (line 367).

SCFAs specification need to be cited in the legend (figure 3).

>We changed to new figure3 and legend.

Reviewer 2 Report

I congratulate you for your excellent work and methodology. Nevertheless I believe that scarcity of participants affected by constipation (Only three participants in MIU group vs 2 in control group), does not allow you to assure that:"MIU improved constipation", so you can not present  those results as a conclusion in your abstract. 

Author Response

Reviewer 2

I congratulate you for your excellent work and methodology. Nevertheless, I believe that scarcity of participants affected by constipation (only three participants in MIU group vs 2 in control group), does not allow you to assure that “MIU improved constipation”, so you can not present those results as a conclusion in your abstract.

Our comments:

>Thank you for your evaluation to our manuscript.

We revised the abstract according to your suggestion as follows; MIU intake reflected the intestinal environment through increased production of three major SCFAs and slgA, and accelerated daidzein-to-equol metabolic conversion, suggesting the beneficial health effects of MIU” (line 31-33)

We appreciate for your consideration.

Round 2

Reviewer 1 Report

The authors improved the article based on the review carried out. However, there are still several concerns that need to be changed.

ABSTRACT:  to remove

  • Primary and secondary outcome (Line 25 and 27). This specification is poorly enclosed. In addition, line 86-89 must also be excluded.
  • “ of three SCFAs” (line 28, 29) repetition of the specification included in line 27.
  • “The conversion efficiency was associated with abundance of Bacteroides ovatus (line 32). If this study did not have enough data for multiple logistic regression analysis, how was this abundance of Bacteroides measured?

INTRODUCTION: I don't know what ‘body maintenance’ would be (line 73)… The objective is not well described (there is no good text, such as clarity, concision and logical sequence). Predominantly, the purpose of the present study was to assess whether DSW-based drinking water (MIU; hardness) could modulate intestinal microbebiomarkers in healthy adults.

MATERIAL AND METHODS:

  • Line 76-78 (added) must be deleted. This information is already described on line 81-83
  • It is highly desirable that the data presented have a normal statistical representation. Why did the table 1 not show the data dispersion as well as the figures? If the data are parametric, the mean followed by the standard deviation (SD) or standard error mean (SEM) must be used.

Examples: BMI (kg/m2): 22.6 ± 0.8

If the data is non parametric, use the median (IQT)

Example: Age (year): 68 (63‒73)

*These data must not be omitted

Line 96-98 must also include the mean (if parametric) ± SD or SEM.

  • According to the authors “There were no significant differences in terms of age, sex, body mass index, and biomarker concentrations between the two groups (Mann–Whitney U test).” (line 99-100). If  Mann-Whitney U test was used, it is because all data are non-parametric, such as line 253 for table 2… Which test was used to assess normality? Authors in the line 125 state “Nonparametric analysis was conducted to assess the differences in these biomarkers before and after the intervention.” In statistic, normality tests are used to determine if the data are wellmodeled by a normal distribution. After tests such as Kolmogorov-Smirnov, Shapiro-Wilk or others. 
  • Line 201, “Throughout the study, we basically performed statistical analysis with nonparametric analyses unless otherwise indicated whenever necessary”. It would be more accurate to consider that thenormal distribution of continuous variables was evaluated using the … test. Data in each group were compared by analysis of variance (to parametric data) or Kruskal-Wallis tests (if non parametric), followed by the tests… respectively for data in the case of a simple comparison. *At no time did the authors describe that ANOVA was performed for parametric data. As well as post-tests were not mentioned.

RESULTS

  • Table 1 and Table 2 “Characteristics of participants in preintervention of two groups (average). median)” (line107) needs to be adjusted for Characteristics of participants of the study or Baseline characteristics. Mean or median with dispersion are usually included below the table.
  • Line 128-129, authors affirm that multiple logistic regression analysis was performed. If this has been done, the relationship between the outcome should be found in this article in result’s section.
  • In Figure 2, to change: Differences in the changes to sIgA concentrations throughout the intervention period in the MIU (a) and control (b) groups (average) to Differences in the changes to sIgA concentrations throughout the intervention period in the MIU (a) and control (b) groups.  Usually, in the legend the description should be the data represent the means ± SD.

*The n used does not need to be repeated in the text. They are in the figure. (Line 235).

  • On line 240, cresol should be replaced by indol. The table 2 did not show indol levels were statistically different. Only p-Cresol is increased for the MIU group.
  • Figure 3A did not show that SCFA concentrations increased in the MIU group. A significantly decrease was onlyobserved in the control group considering the study time (*p < 0.1). Furthermore, this p<0.1 is oflittle significance for acceptance. (Line 278). Why figure 3B is shown in bar. These would be parametric data? If yes, the mean must be accompanied by SD or SEM as authors depicted in Figure 2 for IgA. Importantly, the results demonstrated that the levels of the main SCFAs were decreased in the control group. The difference found between MIU treated and control (3B) was due to the decreased levels for control group.SCFAs for MIU group did not change with the treatment.
  • The average changes in equol concentrations (line 296) throughout the intervention period between the two groups presented in Table 2 must be show with MEDIAN and (values in parentheses should indicate the smallest and largest detected in samples).
  • Line 317, it was descripted thatthe efficiency of converting daidzein-to-equol is associated with the abundance of Bacteroides ovatus (p <0.01). It is essential that the odds ratio (95% CI) for the individual variables reported is shown in a tablefor conditional or cluster-specific measures of association or intracluster measures of association. This could be interpreted as having an effect conditional on the random effect being held constant.

DISCUSSION

These fragmentsneedto be reconsidered and/or removed:

  • Line 349, authors state that MIU (hardness, 88) increased the concentrations of the three major SCFAs (acetic, propionic, and butyric). This is not correct. The pre- and post-treatment value has not changed.
  • “Metabolic efficiency of daidzein-to-equol conversion was accelerated with an increase in the proportion of intestinal ovatus in the MIU group” (line 356) was not demonstrated statistically by the multiple logistic regression. Also line 379-383. And line 389 … the increase in the proportion of B. wexleraein the post-intervention period was observed in all five equol producers, suggesting that B. wexlerae influences equol production.” This has not been demonstrated either.
  • Fora previous clinical study [32] reported on line 358, it would be appropriate and significant if authors mentioned that… We previously found…
  • Line 363…“benefit from disadvantage and precisely evaluate these effects in body.” Effects in body is very subjective. New investigations could assess these effects on immune and inflammatory responses, gut microbiota, and microbial products in healthy adults.
  • Line 404 “…increased concentration of indol was observed in postintervention in the MIU group.” The data showed increased levels of p-cresol.
  • Based on evaluated data and result found,  conclusion (line 422-426) is beyond what the investigation revealed. For me, long-term intervention with MIU induced sIgA production and increased the metabolic conversion of daidzein-to-equol, suggesting an adaptation of the host-microbe to influence human health.”

Author Response

Our Comments to reviewer 1

Thank you for your constructive and concrete comments for our manuscript. We described our comments point-by-point as below. We hope that our manuscript would be improved to better.

ABSTRACT:  to remove

  • Primary and secondary outcome (Line 25 and 27). This specification is poorly enclosed. In addition, line 86-89 must also be excluded.

>We described the fact as follow; “We measured fecal biomarkers of nine short-chain-fatty-acids (SCFAs) and secretory immunoglobulin A (sIgA), and urinary isoflavones.” in abstract (line 25-26). According to your suggestion, line 86-89 was deleted.

  • “ of three SCFAs” (line 28, 29) repetition of the specification included in line 27.

>We changed to “MIU intake significantly affected one SCFA (butyric acid)” (line 27-28).

  • “The conversion efficiency was associated with abundance of Bacteroides ovatus” (line 32). If this study did not have enough data for multiple logistic regression analysis, how was this abundance of Bacteroides measured?

>This sentence was removed in abstract, and we modified in manuscript (line 212-213).

INTRODUCTION: I don't know what ‘body maintenance’ would be (line 73)… The objective is not well described (there is no good text, such as clarity, concision and logical sequence). Predominantly, the purpose of the present study was to assess whether DSW-based drinking water (MIU; hardness) could modulate intestinal microbe biomarkers in healthy adults.

>According to your suggestion, we changed to “The purpose of the present study was to assess whether DSW-based drinking water (MIU; hardness 88) could modulate intestinal microbe biomarkers in healthy adults” (line 69-70).

MATERIAL AND METHODS:

  • Line 76-78 (added) must be deleted. This information is already described on line 81-83.

>We deleted that.

  • It is highly desirable that the data presented have a normal statistical representation. Why did the table 1 not show the data dispersion as well as the figures? If the data are parametric, the mean followed by the standard deviation (SD) or standard error mean (SEM) must be used.

Examples: BMI (kg/m2): 22.6 ± 0.8

If the data is non parametric, use the median (IQT)

Example: Age (year): 68 (63‒73)

*These data must not be omitted.

>We evaluated the data dispersion by Kolmogorov-Smirnov, and the data did not show normal distribution. We performed nonparametric analysis (table 1 and 2) and added the IQR in parentheses in the tables.

Line 96-98 must also include the mean (if parametric) ± SD or SEM.

>We performed nonparametric analysis (table 1 and 2) and added the IQR in parentheses in the tables. Thus, we deleted the information and described only n in each group (line 89-90).

  • According to the authors “There were no significant differences in terms of age, sex, body mass index, and biomarker concentrations between the two groups (Mann–Whitney U test).” (line 99-100). If Mann-Whitney U test was used, it is because all data are non-parametric, such as line 253 for table 2… Which test was used to assess normality? Authors in the line 125 state “Nonparametric analysis was conducted to assess the differences in these biomarkers before and after the intervention.” In statistic, normality tests are used to determine if the data are well modeled by a normal distribution. After tests such as Kolmogorov-Smirnov, Shapiro-Wilk or others. 

>We apologize for your confusion. We forgot to show the IQR when we used nonparametric analysis such as tables mentioned above. Furthermore, according to your suggestion, normality test was already performed by Kolmogorov-Smirnov, indicating that we could not find normal distribution. As you know, basically nonparametric analysis is considered as valuable analysis with high robustness and used widely in general. Thus, we used nonparametric analysis. We described these in the “2.7 Statistical analysis” (line 202-203). We appreciate for your consideration.

  • Line 201, “Throughout the study, we basically performed statistical analysis with nonparametric analyses unless otherwise indicated whenever necessary”. It would be more accurate to consider that the normal distribution of continuous variables was evaluated using the … test. Data in each group were compared by analysis of variance (to parametric data) or Kruskal-Wallis tests (if non parametric), followed by the tests… respectively for data in the case of a simple comparison. *At no time did the authors describe that ANOVA was performed for parametric data. As well as post-tests were not mentioned.

>According to your suggestion, we described as follows; “Normality test was performed by Kolmogorov-Smirnov to assess the normal distribution.” in the “2.7 Statistical analysis” (line 202-203). After that, we eventually used appropriate nonparametric analysis (Mann-Whitney and Wilcoxon) whenever necessary.

RESULTS

  • Table 1 and Table 2 Characteristics of participants in preintervention of two groups (average). median)” (line107) needs to be adjusted for Characteristics of participants of the study or Baseline characteristics. Mean or median with dispersion are usually included below the table.

>We performed nonparametric analysis (table 1 (Mann-Whitney) and 2 (Wilcoxon)) and added the IQR in parentheses in the tables. We corrected these.

  • Line 128-129, authors affirm that multiple logistic regression analysis was performed. If this has been done, the relationship between the outcome should be found in this article in result’s section.

>We apologize for your confusion. We already described these in result’s section (line 268-269). 

  • In Figure 2, to change: Differences in the changes to sIgA concentrations throughout the intervention period in the MIU (a) and control (b) groups (average) to Differences in the changes to sIgA concentrations throughout the intervention period in the MIU (a) and control (b) groups.  Usually, in the legend the description should be the data represent the means ± SD.*The n used does not need to be repeated in the text. They are in the figure. (Line 235).

> Thank you for your constructive comments. According to your suggestion, we revised these in legend and deleted the sentence concerned with “n” in line 237.

  • On line 240, cresol should be replaced by indol. The table 2 did not show indol levels were statistically different. Only p-Cresol is increased for the MIU group.

>I am sorry for the mark checked in table 2. Indol levels were statistically different. The sentence (line 241) is correct. We corrected the position of the mark in table 2.

  • Figure 3A did not show that SCFA concentrations increased in the MIU group. A significantly decrease was only observed in the control group considering the study time (*< 0.1). Furthermore, this p<0.1 is of little significance for acceptance. (Line 278). Why figure 3B is shown in bar. These would be parametric data? If yes, the mean must be accompanied by SD or SEM as authors depicted in Figure 2 for IgA. Importantly, the results demonstrated that the levels of the main SCFAs were decreased in the control group. The difference found between MIU treated and control (3B) was due to the decreased levels for control group. SCFAs for MIU group did not change with the treatment.

>Thank you for your suggestion. We understand what you mean. Thus, we deleted “significantly” and changed in legend (line 280) and appropriate positions in the manuscript (line 261). We performed nonparametric analysis with Wilcoxon, showed the bar (whisker) in Figure 3A. On the other hand, figure 3B shows the percentage for population of responders among participants in two groups, analyzed by chi-squared test, and described these in appropriate position (line 262-265). We appreciate for your understanding.

  • The average changes in equol concentrations (line 296) throughout the intervention period between the two groups presented in Table 2 must be show with MEDIAN and (values in parentheses should indicate the smallest and largest detected in samples).

>We performed nonparametric analysis (table 1 and 2) and added the IQR in parentheses in the tables.

  • Line 317, it was descripted that the efficiency of converting daidzein-to-equol is associated with the abundance of Bacteroides ovatus (<0.01). It is essential that the odds ratio (95% CI) for the individual variables reported is shown in a table for conditional or cluster-specific measures of association or intracluster measures of association. This could be interpreted as having an effect conditional on the random effect being held constant.

>As you pointed out previously, multiple logistic regression analysis performed to evaluate the relationship between the daidzein-to-equol conversion efficiency and relative abundance of bacteria may not be rigorous evidence. Thus, we changed to the following sentence. “Thus, relative abundance of equol-producing bacteria detected in the equol producers was compared before and after drinking MIU. Of 15 equol-producing bacteria, the median of relative abundance of Bacteroides ovatus especially increased from 0.064% (IQR, 0.021-0.262%) to 0.126% (0.034-0.217%) without statistically significance (Wilcoxon)” (line 317-321). Furthermore, we modified the context in the appropriate position in the manuscript as follows; “Of 15 equol-producing bacteria detected in 32 subjects, the relative abundance of B. ovatus especially increased in the MIU group without statistically significance, suggesting that MIU intake influenced the equol-producing bacteria including B. ovatus” (line 379-382). We appreciate for your understanding.

DISCUSSION

These fragments need to be reconsidered and/or removed:

  • Line 349, authors state that MIU (hardness, 88) increased the concentrations of the three major SCFAs (acetic, propionic, and butyric). This is not correct. The pre- and post-treatment value has not changed.

>As you remember, in the first manuscript, we showed that the average of total amount of the 3 SCFAs slightly increased in MIU group but not in control group (decreased). The change through the intervention in MIU group did not show the statistical difference. Of course, as you pointed out above, we understand that the difference found between MIU treated and control was due to the decreased levels for control group. Thus, we changed to “In this study, the average of total amount of the three SCFAs slightly increased in MIU (hardness, 88), however, MIU mainly increased the sIgA production as fecal biomarker.” (line 350-352). We appreciate for your understanding.

  • “Metabolic efficiency of daidzein-to-equol conversion was accelerated with an increase in the proportion of intestinal ovatus in the MIU group” (line 356) was not demonstrated statistically by the multiple logistic regression. Also line 379-383. And line 389 … the increase in the proportion of Bwexlerae in the post-intervention period was observed in all five equol producers, suggesting that Bwexlerae influences equol production.” This has not been demonstrated either.

>Regarding to the efficiency and B. ovatus, we changed in appropriate positions as mentioned above.

>Regarding to line 389 concerned with Bwexlerae, we described (line 326-328) and showed the list (table4) for increased bacteria detected in postintervention of 5 equol-producers, whose number is not adequate for statistical analysis. However, we found the relative abundance of these bacteria detected in 5 equol-producers in postintervention. We think that the list (bacteria increased in 5 equol-producers who became equol producers during the intervention period) is necessary/helpful information for considering new equol-producing bacteria among intestinal microbes. We attenuated the description and changed to “The relative abundance of B. wexlerae in the postintervention period was observed in all five equol producers, suggesting that B. wexlerae could be involved in equol production.” in Discussion (line 390-392). We appreciate for your understanding.   

  • For a previous clinical study [32] reported on line 358, it would be appropriate and significant if authors mentioned that… We previously found…

>Thank you for your suggestions, we modified as follows; These findings were not observed in previous clinical study with RDSW [32]. We previously found increased concentrations of five SCFAs in the RDSW. Thus, the influences…. (line 358-360)

  • Line 363…“benefit from disadvantage and precisely evaluate these effects in body.” Effects in body is very subjective. New investigations could assess these effects on immune and inflammatory responses, gut microbiota, and microbial products in healthy adults.

>Thank you for your suggestion, we changed to “New investigations could assess these effects on immune and inflammatory responses, gut microbiota, and microbial products in healthy adults” (line 363-365).

  • Line 404 “…increased concentration of indol was observed in postintervention in the MIU group.” The data showed increased levels of p-cresol.

>I am sorry for the mark checked in table 2. Indol levels were statistically different. The sentence (line 241-242) is correct. We corrected the position of the mark in table 2.

  • Based on evaluated data and result found, conclusion (line 422-426) is beyond what the investigation revealed. For me, long-term intervention with MIU induced sIgA production and increased the metabolic conversion of daidzein-to-equol, suggesting an adaptation of the host-microbe to influence human health.”

>Thank you for your suggestion. According to your comments, we described as follows; This clinical study revealed that a long-term intervention with MIU mainly induced sIgA production and increased the metabolic conversion of daidzein-to-equol, suggesting an adaptation of the host-microbe to influence human health. (line 425-427).
